# Preferences of women in difficult life situations for a physical activity programme: protocol of a discrete choice experiment in the German NU-BIG project

Sara Pedron [1], Annika Herbert-Maul [2], Alexandra Sauter,[3] Stephanie Linder,[2] Raluca Sommer,[2] Markus Vomhof,[4,5,6] Veronika Gontscharuk,[4,5,6] Karim Abu-Omar,[2] Ansgar Thiel,[7] Heiko Ziemainz,[2] Rolf Holle,[8] Michael Laxy[1,5,9]

For numbered affiliations see end of article.

**Correspondence to**
Dr Sara Pedron;
sara.pedron@tum.de

## ABSTRACT

**Introduction** The BIG project ('Bewegung als Investition in die Gesundheit', ie, 'Movement as Investment in Health') was developed in 2005 as a community-based participatory research programme to offer accessible opportunities for physical activity to women in difficult life situations. Since then, the programme has been expanded to eight sites in Germany. A systematic evaluation of BIG is currently being conducted. As part of this effort, we strive to understand the preferences of participating women for different aspects of the programme, and to analyse their willingness to pay.

**Methods and analysis** In this protocol, we describe the development and analysis plan of a discrete choice experiment (DCE) to investigate participants' preferences for a physical activity programme for women in difficult life situations. The experiment will be embedded in a questionnaire covering several aspects of participation in the programme (eg, reach, efficacy and further effects) and the socioeconomic characteristics of all active participants. After a thorough search of the literature, BIG documents review and expert interviews, we identified five important attributes of the programme: course times, travel time to the course venue, additional social activities organised by BIG, consideration of wishes and interests for the further planning of courses and costs per course unit. Thereafter, we piloted the experiment with a sample of participants from the target group. After data collection, the experiment will be analysed using a conditional logit model and a latent class analysis to assess eventual heterogeneity in preferences.

**Ethics and dissemination** Understanding women's preferences will provide useful insights for the further development of the programme and ultimately increase participation and retention. The questionnaire, the included DCE and the pretest on participants received ethical approval (application no. 20-247_1-B). We plan to disseminate the results of the DCE in peer-reviewed journals, national conferences and among participants and programme coordinators and organisers.

## STRENGTHS AND LIMITATIONS OF THIS STUDY

⇒ This discrete choice experiment (DCE) is the first to evaluate participants' preferences for cooperative planning and social inclusion aspects of a community-based participatory research programme in the physical activity promotion field.
⇒ This DCE is the first to target preferences for physical activity opportunities in a vulnerable group, namely women in difficult life situations.
⇒ This DCE necessarily represents a simplification of the decision-making process of women participating in the programme, where several other aspects might play a role in driving women's decisions to keep participating.
⇒ As the target group consists of women who already participate in the programme, we will analyse preferences for sustained participation, but will not be able to evaluate preferences for starting anew in the offered physical activity courses.

## INTRODUCTION

Several studies have demonstrated that leisure time physical activity follows a socioeconomic gradient: individuals with a lower education, lower income and lower occupational position were consistently shown to engage less frequently in physical activity than their higher socioeconomic status counterparts.[1–3] In fact, socioeconomic factors are one of the major determinants of cardiovascular and metabolic conditions, which are among the most important drivers of morbidity and mortality worldwide.[4] The effect of socioeconomic factors on cardiometabolic health is multifaceted, but runs mostly via physiological, psychosocial and behavioural factors, with varying degrees of influence in different contexts and life stages.[5] Furthermore, several reviews from different contexts have shown a

gender gap in physical activity, demonstrating that women are more at risk of sedentary behaviour than men because of several barriers, such as family responsibilities, safety concerns or lack of culturally appropriate sport opportunities.[6–10] Therefore, the intersection of socioeconomic factors and sex-specific factors makes women from disadvantaged socioeconomic backgrounds a vulnerable group at serious risk of sedentary behaviour.

The fact that women from disadvantaged socioeconomic backgrounds are especially at risk for physical inactivity has also been demonstrated in the German context.[11–14] Therefore, policies and programmes aiming to reduce barriers to participation (economic, time-related, cultural and social) and increase the amount of leisure time physical activity, especially in socially disadvantaged populations, are needed. Although the majority of programmes involve individual cognitive-behavioural approaches to enhance physical activity,[15] it has been argued that a broader approach also targeting structural aspects might be necessary to improve lifestyles effectively.[16] This is especially the case if such programmes manage to activate already existing assets in the communities and increase the offer of targeted possibilities, for example by directly considering and overcoming actual and perceived barriers in the targeted groups.[17]

One example of such programmes from the German context is the 'BIG' project ('Bewegung als Investition in Gesundheit'—'Movement as Investment in Health'). The programme is based on a community-based participatory research approach, that is, a research paradigm that aims primarily at closer communication with and involvement of the communities that are the object of the investigation to shape, manage and evaluate the intervention or the context studied.[18] In this case, the programme specifically targets physical activity courses for women in difficult life situations, for example, having a low household income, a migration background, being unemployed, reliant on welfare payments or single mothers.[19 20] The programme has since expanded to a further 20 communities (eight of them are still active). In each of these, a 'cooperative planning approach' was adopted, where women from the target group were explicitly and regularly included in the planning and organisation of the courses, with the aim of overcoming barriers and increasing participation and retention.[21–26]

Even though the project strives to include women from the target group in the planning and organisational phase, little is known about their preferences. Understanding and evaluating the preferences of participating women is important in order to develop the programme further, make it more diversified to reach specific groups of the population and encourage participation over a longer period of time. In order to elicit patient or participant preferences, economic stated-preferences methods such as discrete choice experiments (DCEs) have been used increasingly in healthcare contexts.[27] This preference elicitation method relies on the economic theory of demand[28] and on random utility theory.[29 30] According

to these theories, when confronted with a choice, individuals will choose the option with the highest level of utility, that is, the option whose characteristics combined generate the highest utility value for the individual. DCE is a choice-based method where individuals are confronted with a series of hypothetical choice scenarios to simulate realistic choices. Respondents have to trade-off one option with all its characteristics against a second option, finally indicating which one of the two they prefer. By evaluating the final choice pattern, the method allows the computation of the relative importance of attributes. If an attribute representing costs is also assessed, then the willingness to pay (WTP) for levels can be elicited, that is, how much on average a participant is willing to pay for an improvement in one attribute.[31]

### Aims of this protocol and of the DCE

Using a DCE, we aim to understand and quantify the preferences of participating women for specific structural characteristics of the BIG programme and to investigate how they differ with respect to sociodemographic and cultural backgrounds. The aim of the present protocol is to outline the preparatory steps for the DCE and to present the final version of the experiment used in the questionnaire. This DCE is the first to evaluate the cooperative planning aspect and social activities component as structural characteristics of a community-based participatory research programme in the physical activity promotion field. Investigating preferences for the BIG physical activity courses will help to deduce possible reasons why women from disadvantaged social backgrounds participate, using a tool that mimics real life choices including a limited offer of alternatives and potential trade-offs. Furthermore, the findings will also help to develop the programme further at the active BIG sites and help to achieve a greater involvement of participating women by aligning the programmes to preferences and thus fostering their integration and the benefits to women from participation. Finally, identifying components that are essential or highly valued by the target group can be useful in informing policymakers to design better strategies for physical activity promotion for women in difficult life situations.

## METHODS AND ANALYSIS

In this section, we explain the background and context in which the DCE is embedded ('The BIG project and The NU-BIG evaluation and the target group'), the preparatory steps that were carried out to set up the DCE ('Development of the DCE to assess participant preferences'), the planned analyses ('Data analysis'), patient involvement activities ('Patient and public involvement') and anticipated strengths and limitations of the study ('Strengths and limitations').

### The BIG project

The BIG project ('Bewegung als Investition in Gesundheit', ie, 'Movement as Investment in Health') was

started in 2005 by the Friedrich-Alexander-Universität Erlangen-Nürnberg (FAU), Germany. The project specifically strives to increase physical activity among women in difficult life situations.[19] Since its start, the programme has been transferred to another 20 communities/municipalities, mostly but not only in Bavaria (Germany).[21] Currently, the programme is being actively implemented at eight sites, with three sites in the starting phase. In the remaining nine sites, the programme was stopped because of different implementation and sustainability issues in the local communities.

The target groups of the BIG programmes are women in difficult life situations, which include women with a low income, a migration background, unemployed women, women relying on welfare payments or single mothers. However, BIG courses are open to all women who decide to participate.

Using a 'cooperative planning approach',[22–25] the programme also aims to involve women from the target group in the planning, organisation and evaluation stages of the project. In this way, highly tailored exercise programmes could be constructed by eliminating barriers (eg, by offering childcare possibilities during the course and by offering courses that respect religious norms, etc). Ultimately, the programme aims to maximise the participation and inclusion of women, thereby achieving broader empowerment, with potential effects on self-efficacy and other health behaviours (eg, healthy diet and smoking). Furthermore, this approach facilitates the inclusion of other women from their social network with a similar socioeconomic background, fostering integration and social exchange within the programme itself.

At each site, the programme offers several courses, in both gyms/halls (fitness courses) and swimming pools (water courses). The fitness courses include, for example, Pilates, yoga and aerobics. The water courses include water fitness and swimming courses. Furthermore, at some sites, additional social and cultural activities are organised, such as a 'breakfast for women' ('Frauenfrühstück') several times per year, cook-in evenings with healthy dishes from different cultures and further occasions for mutual exchange. Given the participatory nature of the programme and the active role that women have in shaping its content and forms, the courses offered and additional activities differ from site to site (eg, workshops for conflict management and stress reduction or self-defence).[21] A detailed overview of the types of courses offered at each active site in the last year before the pandemic (2019), the number of participants in each course and the additional activities can be found in online supplemental appendix 1.

For the success of the programme, women from the target group should be reached by these offers and should agree to participate. Furthermore, another very important aspect is their continued participation, as only sustained participation leads to health improvements. Therefore, understanding the factors that drive their sustained participation (eg, via a DCE) is of paramount importance.

### The NU-BIG evaluation and the target group

The present DCE is part of a general effort to evaluate the impact of the BIG programme, called the NU-BIG study ('Nachuntersuchung des BIG-Projekts'), funded by the German Federal Ministry of Education and Research. The evaluation will follow the RE-AIM framework and was described in detail in the study protocol for the NU-BIG project.[32] The evaluation focuses on investigating the long-term effects of BIG on health behaviour, inclusion and social participation, as well as factors facilitating long-term implementation and transfer. The study consists of a cross-sectional evaluation using a mixed-methods approach, including questionnaires for participating women, qualitative interviews for local organisers and focus groups with participating women.

The present DCE will be part of the questionnaire, which will be distributed to all women who participate in BIG courses between May 2022 and winter 2022/2023 at active sites (excluding sites in the starting phase). Based on prepandemic participation rates, it was estimated that the study population will amount to around 800 participating women at eight sites. The questionnaire will be administered as a 'paper and pen' survey, by handing the paper questionnaire to the target group in person during the BIG courses. Owing to the diversity of BIG activities at each site, an adjustment of the mode of administration will be required for several courses. In consultation with the trainers for each course, we will adapt the way we distribute the questionnaire to the specific requirements of each course (eg, outdoor courses and swimming classes) and to the specific linguistic background of participating women by making the questionnaire available in German, Arabic, English, Russian and Turkish.

### Development of the DCE to assess participant preferences

To derive preferences, it is important to carefully develop and then transfer the DCE into an experimental design following methodological standards.[29 33] The DCE is a type of conjoint analysis, that is, a stated-preference method that involves an indirect comparison of choices, for example, via ranking, rating or choice designs, to evaluate and quantify preferences for several attributes of an intervention.[33] Therefore, designing a DCE involves a thorough identification of attributes and levels that define the intervention, and the construction of choice sets within the experimental design. In the next paragraphs, we describe this process in detail in the context of the NU-BIG study.

### Problem and target group definition

The main aim of the DCE is understanding which aspects of the programme favour sustained participation in the programme and help to keep participating women interested in new courses.

The target group consists of all women who are actively participating in BIG courses, as the questionnaire will be handed in during classes starting from May 2022 until winter 2022/2023.

### Identification of attributes and levels and transfer to the experimental design

The DCE will be embedded in the questionnaire and will present women with a series of hypothetical comparisons between two physical activity courses with predefined characteristics, asking them to choose which option they like the most.

In a first step, all programme characteristics (attributes) that might influence individual preferences for participation in BIG courses should be identified. At the same time, appropriate levels of these attributes should be determined. All included attributes should be relevant for the choice of course, should be easily influenced and close to reality, should have a substitutive relation and should not be dominant.[34] To identify attributes and their relative levels, we followed a systematic approach. We describe all steps below.

#### Compilation of evidence (literature search)

The literature was searched for physical activity programmes/interventions that were evaluated using a DCE. The full search strategy, flow chart and results can be browsed in online supplemental appendix 2. We identified 19 studies, for which we systematically retrieved attributes and corresponding levels. We divided these attributes into meaningful groups: exercise frequency and duration, context, type of exercise, outcomes and goals, schedule, costs, travel time and location, support and other. It is important to note that no study specifically included women exclusively as a target group.

#### Compilation of evidence (programme specific)

We reviewed previous documents from the BIG programme (qualitative and quantitative data, protocol of the kick-off meeting) to identify potential attributes and levels in addition to those identified by the literature search. Each document was searched independently by two researchers from the group (AS, SL, AHM and SP). Resulting attributes and levels were then pooled, discussed by the group (SP, AHM, SL and AS) and compared with the results from the systematic literature search (full list available in online supplemental appendix 2).

In a further step, we selected only attributes that are applicable in the context of the BIG courses and that are already implemented or potentially implementable in the programme. For example, attributes and levels that included mixed classes with men or individual courses were excluded as BIG only offers group courses for women. Furthermore, attributes involving the presence of a personal coach were also excluded as this is not intended or feasible in BIG. Additionally, attributes referring to programmes targeting specific illnesses were also excluded.

This selection led to a list of 10 potentially important attributes (online supplemental appendix 3), which served as the basis for discussion with the experts in the following step.

#### Consultation with experts

This step involved an online discussion of the previously identified list of 10 attributes and their respective levels with five 'BIG experts'. They are BIG course instructors, organisers or long-term participants, who actively took part in shaping BIG courses at their site and who are routinely interviewed and involved in the planning phases of the NU-BIG evaluation. The expert group of women was created at the beginning of the NU-BIG evaluation to implement the grounding principles of community-based participatory research and includes highly motivated women who are active at their sites in different roles. They receive a small compensation payment for their participation and are involved in multiple projects within the NU-BIG evaluation.

In the meeting, which was headed by SP, AHM and AS, we explained to the experts the goals of the DCE, and we presented them with a first example of trade-off decisions within a fictitious choice set. After this, each attribute and the respective level was thoroughly discussed in the group. Thereby, we specifically asked whether the 10 reported characteristics are relevant ones and whether they represent important drivers in the decision to participate. Furthermore, we discussed their clarity and understandability, and we explicitly asked whether some important aspects might be missing. At the end of the meeting, we also asked the five experts to rank the attributes based on the perceived relevance for participating women with an interactive tool to spark further discussion.

The discussion led to interesting insights into the reasons for participating from the point of view of the course instructors and organisers. Following this discussion, one additional attribute was identified as an important one ('flexible intensity of the course') and two others were reframed in a different way. We report the results of this discussion, including the new list of attributes and levels, in online supplemental appendix 3.

Based on these insights, the list of 11 attributes was analysed further by researchers (SP, AHM and MV). In our screening of the attributes, we ensured that all attributes were relevant for the choice of course and were easily influenced within the BIG programme. Furthermore, we also ensured that they were in a substitutive relation (ie, the worsening of one attribute can be compensated by an improvement in another attribute) and were not dominant (ie, no attribute should be a necessary criterion for or against participation).[34] The detailed decision steps and the resulting list of attributes and levels are reported in online supplemental appendix 3. The resulting list includes five attributes, that is, (1) course time, (2) travel time to the BIG course, (3) additional activities organised by BIG, (4) consideration of wishes or interests and (5) costs per course unit. Based on discussion with the

experts, we decided to keep some attributes fixed, that is, to introduce an initial statement clarifying that all courses would have specific characteristics. These include the presence of childcare, the presence of trial lessons, course duration and the framework (eg, the hall is not visible from the outside). These fixed attributes are in fact usual components of all BIG courses that are already implemented to reduce barriers to participation.

### Pretest with participants

Based on the selected attributes and levels, a preliminary experimental design was conducted to prepare a full DCE for a pretest. We tested the full DCE questionnaire with a set of voluntary participants from the target group. To this end, we recruited participating women and performed semistructured qualitative interviews using the think aloud technique based on a topic guide until a saturation of new content was achieved.[35 36] This led us to include six women from two BIG sites (Erlangen and Nürnberg). The interviews were carried out by telephone and were recorded after asking for the consent of the participants. The interviewee was asked to read a paper version of the questionnaire that was previously sent by mail, and loudly report her thoughts and choices. The notes and recordings of the phone call were then revised by the interviewer (SP) and by a second researcher (MV), in order to identify the main problems and any improvements necessary.

We are aware that both samples selected for the expert interviews and the pretest might not be representative of the targeted population of participating women. However, it was impossible for the study team to directly access participants from the study population (the study was planned before late spring 2022, when only selected courses were taking place because of COVID-19 restrictions). The potential bias induced by this issue will be considered in the final evaluation.

In the pre-test with participants, we paid particular attention to the following criteria:
1. Are all the most important attributes included in the DCE?
2. Is one or more of the included attributes dominant?
3. Are all levels appropriate?
4. Are the task and the questionnaire understandable and feasible?

The interviews followed a prespecified outline and were recorded for quality assurance and analysed using content analysis methods by two researchers (SP and MV).[37] This step led to a general confirmation of the understandability of the experiment among women with different backgrounds, linguistic skills and from different cities. The task was understood properly, and all women engaged in the trade-offs proposed in the DCE with ease. Drawing from the feedback of participating women, we made some adjustments in missing context information (eg, the courses take place only during the week, the number of appointments was specified). Furthermore, we modified the range of costs per training unit proposed in

**Table 1** Attributes and levels included

| Attributes | Levels |
| --- | --- |
| Course time | Early morning (8–10 am) |
| | Morning (10 am–12 pm) |
| | Afternoon (4–6 pm) |
| | Evening (6–8 pm) |
| Travel time to the BIG course | Maximum 10 minutes |
| | Maximum 20 minutes |
| | Maximum 30 minutes |
| Additional activities by BIG | No additional social activity is organised |
| | Additional social activities are organised |
| Consideration of wishes and interests for the further planning of courses | Interests/wishes are not asked/considered |
| | Interests/wishes are asked/considered |
| Costs per course unit | €2.00 |
| | €3.50 |
| | €5.00 |
| | €6.50 |
| | €8.00 |
| | €9.50 |

The original version in the German language is available in online supplemental appendix 4.
BIG, Bewegung als Investition in Gesundheit—Movement as Investment in Health.

the DCE. Initially, we considered a price range between €0 and €5 per training unit, with €1 intervals (the levels were: €0, €1, €2, €3, €4 and €5). However, the price was perceived as an irrelevant attribute by the participants, as even the highest level was considered very affordable and comparable with the levels below. Based on specific indications of participating women on the price of courses that they currently participate in (up to €8 per course unit), we decided to increase the price range to €9.5. Furthermore, we received mixed feedback regarding including the level €0: although some women perceived this as very advantageous, others interpreted this as a sign of poor quality. As we assume linearity in the cost attribute, we decided to eliminate this level. The final cost levels thus range between €2 and €9.5, with €1.5 intervals.

### Final attributes and levels included

The final list of attributes and levels included in the DCE is reported in table 1 (original language version in online supplemental appendix 4). We included five attributes, with a varying number of levels. The first attribute is the course time, that is, when the course takes place during the day. For this attribute, we considered four levels: early morning, morning, afternoon and evening. The second attribute represents the travel time needed to reach the

course location. This encompasses three levels: maximum 10, 20 or 30 min. The third attribute represents the possibility of organised social activities within the BIG course, such as joint breakfasts (called 'women's breakfasts') or dinners where all participating women are invited to socialise outside of the courses. This attribute has two levels: activities are organised or not. The fourth attribute considers the possibility of listening to participants' interests and wishes, to be considered in the further planning of courses. Also, this attribute is a binary one, that is, either interests/wishes are considered or not. The final attribute is the cost of a course unit, ranging from €2 to €9.5 in €1.5 intervals. Besides a clear preference for lower costs per course unit, a priori we cannot hypothesise which levels will be preferred by women.

### DCE questionnaire design and construction of choice sets

We constructed the final DCE experimental design following best practice guidelines.[38] The number of attributes and levels included in our DCE would generate 288 potential combinations ($2^2 \times 3 \times 4 \times 6$). The attribute costs per course unit entered were considered as a continuous variable in the experimental design. Owing to a decentralised administration of questionnaires, we were not able to implement a block design. As the number of possible choice scenarios is very large, we used a D-efficient design to select the most efficient combination of choice sets, while at the same time allowing the estimation of all the main effects.[27 39]

We employed SAS macros (SAS V.9.4) by Kuhfeld[40] to derive the optimal number of choice sets. The final efficient factorial fractional design (D-efficiency=5.8409, relative D-efficiency=73.0117, D-error=0.1712) consisted of eight unique choice tasks.

All attributes and level combinations were checked for plausibility. In order to test the reliability of the choices, we repeated one item (choice set 4), leading to nine choice sets included in the final DCE. Participants are asked to choose between course A and course B. To avoid increasing the cognitive effort for participants, we did not include any opt-out option.

### The DCE questionnaire

The DCE will be included in the NU-BIG paper-based questionnaire, after a general questionnaire introduction and questions regarding health behaviour, health status, self-efficacy and integration and sociodemographic aspects. The DCE includes one page of detailed information on the content of the DCE. Participants are asked to choose from two options the one they prefer most, that is, the one they would be less likely to reject. An example of a choice set is presented in figure 1, and the full questionnaire in the original language is available in online supplemental appendix 5.

### Data analysis

#### Planned analyses

The included population will be described with respect to their sociodemographic characteristics drawing from the information collected in the NU-BIG questionnaire.

Respondents with missing information in more than half of the choice sets (>4) will be excluded from the analysis. All other women will be eligible for the analysis.

In order to estimate the preference weight of each level within each attribute, we will analyse the DCE data using a conditional logit model. Thereby, we will calculate the relative importance of attributes, using an effects-coding procedure.[41] In a further step, we will express the preference weights as money equivalents (ie, as the marginal rate of substitution between the attributes and the cost attribute), in order to compute the marginal WTP for a change in the level of the attributes.[31]

The target population of women who participate in BIG courses is likely to be a heterogeneous one, especially with regard to sociodemographic characteristics, which might impact preferences (eg, presence of children in the household, migration background and linguistic barriers, religious affiliation, etc). First, we will describe how the

|  | **Class A** | **Class B** |
|---|---|---|
| **Course time** | Morning (10 am - 12 pm) | Early morning (8 - 10 am) |
| **Travel time to the BIG course** | Maximum 20 minutes | Maximum 10 minutes |
| **Additional activities by BIG** | No additional social activity is organised | Additional social activities are organised |
| **Consideration of wishes and interests for the further planning of courses** | Interests/wishes are asked/considered | Interests/wishes are not asked/considered |
| **Cost per course unit** | 8.00 € | 5.00 € |
| **Which class do you prefer? (Please tick the corresponding box)** | ☐ | ☐ |

**Figure 1** Example of choice set. Own translation from German. BIG, Bewegung als Investition in die Gesundheit, that is, Movement as Investment in Health.

**Table 2** Sociodemographic variables used in the latent class analysis

| Sociodemographic characteristics | Measurement | Type of variable | Range/categories |
|---|---|---|---|
| Age | Self-reported | Continuous | Unknown |
| Employment status | Self-reported | Categorical | Employed, retired, unemployed, other |
| Education (highest qualification) | Self-reported | Categorical | No school diploma, mandatory school diploma, middle school diploma, high school diploma |
| Education (school years) | Self-reported | Continuous | Unknown |
| Household monthly income | Self-reported | Categorical | Below €500/month up to €6000/month (in €500 steps) |
| Number of children (below 14 years of age) | Self-reported | Continuous | Unknown |
| Living alone/with partner | Self-reported | Categorical | Living alone vs with partner |
| Health status | Self-reported | Categorical | 1 (very good) – 5 (very bad) |
| Migration status | Self-reported | Categorical | German born vs foreign born |
| Religious affiliation | Self-reported | Categorical | Evangelical church, Catholic church, Islam, Judaism, Russian Orthodox church, no religious affiliation, other |

surveyed population (ie, women who participate in BIG courses between summer 2022 and winter 2022/2023) coincides with the intended target population of women in difficult life situations targeted by the BIG programme. Second, we will carry out a latent class analysis in order to explore the underlying heterogeneity and identify classes of women with different preferences. We will also analyse whether these classes will be different according to age, employment status, education, income, number of children, marital status, health status, migration status and religious affiliation (see table 2 for details).[31] All analyses will be conducted using R-studio.[42]

### Sample size calculation
All participating women at the active BIG sites will be offered the chance to participate in the NU-BIG survey. Therefore, we will have limited possibilities to influence the sample size for the DCE. Nevertheless, computing the minimum required size of the sample, also taking into account different response rates, is crucial for ensuring the feasibility of the DCE.

In order to compute the minimum sample size, we used the following rule of thumb:[43 44]

$$n > \frac{1000\,c}{t \times a}$$

Here, n is the number of participants, c is the highest number of levels (in our case, c=4), t is the number of choice tasks (in our case, t=8) and a is the number of alternatives per choice task (in our case, a=2). In our case, this formula yields a minimum of 250 participants.

Based on participation prior to the COVID-19 pandemic, the number of participants in the BIG courses was estimated to be around 800. However, the context and participation are likely to be different now, so that a conservative approach for the estimated target group is necessary. Even assuming a 50% reduction in the sample, or in the response rate, the sample size is still higher than the minimum suggestion of 250.

### Patient and public involvement
The BIG project is carried out using a 'cooperative planning approach',[19] where women from the target group are involved in the planning and implementation of physical activity courses. Furthermore, in the NU-BIG project, as a comprehensive evaluation of BIG, the point of view of women from the target group was always taken into account by regularly involving the group of 'BIG experts' in every step of the project for targeted feedback.[21 32]

Concretely, the idea for the DCE sparked from the meetings with the 'BIG experts' as a way of formally investigating the preferences of participating women. Both the 'BIG experts' and women from the target group of participants were involved in setting up the DCE in the pretest phases. Their helpful critical feedback was used to further develop the instrument.

### Strengths and limitations
The described DCE is the first to attempt to understand participants' preferences for structural characteristics of a community-based participatory research programme in the physical activity promotion field. Furthermore, it is the first DCE targeting the preferences of women in difficult life situations, a high-risk group when it comes to prevention efforts.

However, we anticipate the presence of some limitations. First, a DCE is based on the most important attributes relevant in the decision-making process of the actors. The trade-off between the number of attributes and the complexity of the choice task may result in the omission of attributes that are still relevant for some participants. Nonetheless, due to the thorough development and testing of the DCE, we expect that the

experiment includes the most relevant attributes for the majority of participants.

Second, as recruitment of the experts and women participating in the pretest were convenient samples, there might be a selection bias for these groups.

Third, we might face a further selection bias as we are not able to measure the preferences of women who left the courses. The preferences of this group might be different from those of women who are still participating in the course. Considering the preferences of women quitting the courses might have led to very interesting insights into the reasons for quitting and which course components might have helped to sustained participation. Therefore, the interpretation of our results is limited to the preferences of women who are still participating in the course.

## ETHICS AND DISSEMINATION

The NU-BIG project was approved by the Ethics Commission of the Friedrich-Alexander-University of Erlangen-Nurnberg (application no. 20-247_1-B). The DCE described in this protocol was included as part of the NU-BIG project and described in detail, also including recruitment for the pre-test with women from the target group.

All women participating in the pretest read an extensive information sheet regarding the DCE and gave written informed consent and received an incentive (€20). All participants in the final NU-BIG paper questionnaire will receive an information sheet and will be required to sign an informed consent form.

The DCE questionnaire and the relative data analysis will be carried out following the principles of good scientific practice.[31 38] We will compile and attach to the final publication the checklist for conjoint analysis applications in healthcare.[33] The results will both be disseminated in academic journals and conferences, and also among participants groups and organising bodies, with the explicit aim of improving the courses offered by BIG at all sites.

**Author affiliations**
[1]Professorship of Public Health and Prevention, Department of Sport and Health Sciences, Technical University of Munich, Munich, Germany
[2]Department of Sport Science and Sport, Friedrich-Alexander-Universitat Erlangen-Nurnberg, Erlangen, Germany
[3]Department for Epidemiology and Preventive Medicine, Medical Sociology, University of Regensburg, Regensburg, Germany
[4]Institute for Health Services Research and Health Economics, Centre for Health and Society, Medical Faculty and University Hospital Düsseldorf, Heinrich-Heine-University Düsseldorf, Düsseldorf, Germany
[5]German Center for Diabetes Research, Neuherberg, Germany
[6]Institute for Health Services Research and Health Economics, German Diabetes Center (DDZ), Leibniz Center for Diabetes Research, Heinrich-Heine-University Düsseldorf, Düsseldorf, Germany
[7]Institute of Sports Science, Social and Public Health Sciences and Interfaculty Research Institute for Sport and Physical Activity, Eberhard Karls Universität Tübingen, Tübingen, Germany
[8]Institute for Medical Informatics, Biometry and Epidemiology, Ludwig Maximilian University of Munich, Munich, Germany
[9]Institute of Health Economics and Healthcare Management, Helmholtz Center Munich, German Research Center for Environmental Health (GmbH), Munich, Germany

**Contributors** SP and ML identified the research question. SP, MV and VG planned the described discrete choice experiment (DCE) in all phases. AH-M and RS worked as coordinators of the whole NU-BIG (Nachuntersuchung des-Bewegung als Investition in die Gesundheit, that is, Movement as Investment in Health) project and contributed to the organisation of the preparatory steps in the DCE. SP, AH-M, AS and SL carried out the necessary steps for the identification of potential attributes and interviewed the experts. SP and AS carried out the pre-test with participants. SP wrote a first draft of the manuscript, and all authors read, commented on and approved the manuscript. KA-O, HZ, AT, AHM and RH were responsible for the funding application and work as coordinators for the NU-BIG evaluation. They contributed expert knowledge to the identification of the research question, the setting up of the DCE and the final manuscript.

**Funding** This work was supported by the German Federal Ministry of Education and Research (grant no. 01EL20122D).

**Competing interests** None declared.

**Patient and public involvement** Patients and/or the public were involved in the design, or conduct, or reporting or dissemination plans of this research. Refer to the Methods section for further details.

**Patient consent for publication** Not required.

**Provenance and peer review** Not commissioned; externally peer reviewed.

**ORCID iDs**
Sara Pedron http://orcid.org/0000-0002-8127-573X
Annika Herbert-Maul http://orcid.org/0000-0001-5372-9952

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
