## [Reviewer comments · BMJ Open]

ARTICLE DETAILS

TITLE (PROVISIONAL)	Preferences of women in difficult life situations for a physical activity programme: protocol of a discrete choice experiment in the German NU-BIG project
AUTHORS	Pedron, Sara; Herbert-Maul, Annika; Sauter, Alexandra; Linder, Stephanie; Sommer, Raluca; Vomhof, Markus; Gontscharuk, Veronika; Abu-Omar, Karim; Thiel, Ansgar; Ziemainz, Heiko; Holle, Rolf; Laxy, Michael

VERSION 1 – REVIEW

REVIEWER	Tomberge, Vica University of Bern
REVIEW RETURNED	23-Oct-2022

GENERAL COMMENTS	Dear authors and editors, I was delighted to be given the opportunity to review the study protocol of your highly relevant study which can hopefully contribute to the insight on the barriers and preferences of women from socially disadvantaged populations to engage in physical activity and other health promotion programs. Please note that the line numbering was not consecutive. All comments refer to the pages assigned by BMJ (not the page numbers on the authors' document, these differ). First of all, I am not familiar with the DCE, thus I will limit myself to:  • giving feedback on the appropriateness of the conceptual development of the options included (based on contextual development of measures in general) • but I can not give any feedback on the appropriateness of the choice set design (2.3.3. l. p. 13 22 ff.) or the analyses of it p. 14, l. 14 ff. Abstract  • Abstract contains shortforms which are not explained (BIG) and methodological approaches which are not self-explanatory and need a short definition (discrete choice experiment) [ ] Please try to write the abstract sufficiently explicit to understand it on its own. - Also your abstract contains large parts of describing preparatory work (deveelopment of tool) which is a bit confusing since this work was already conducted and is rather a tool for the the experiment your plan, cosider focusing on the experiment itself (see more specific comments on this below) Introduction /Theoretical background (p.5-6)  • Your literature discussed socioeconomic barriers to health and health behavior exclusively, given that you are placing a women focused health program I would expect to read about the particular relevance of women as a target group and the intersection of
---

	gender and socioeconomic status related to health, health behavior and barriers.  • In line 45 p.5 you mention that the program is based on community-based participatory research, can you define/ elaborate this? In line 52 you then mention a “cooperative planning approach” which you define sufficiently, do you mean the same by this? - I am not convinced by the literature you cite (one position statement) for supporting that structural aspects, instead of individual-level determinants, might prove more effective to enhance physical activity. Since this is a big claim, convince me by adding experimental evidence or qualitative studies at least. P. 5 l. 30 ff. . Maybe it is not necessary to state that they are “more” effective, structural aspects are maybe simply one of many components of health behavior (besides individual cognitions etc). - I need clarification on who you mean when you talk about communities, p.5 l. 50, is this a fixed group of people or a specific social group? Or is it the same as “sites” referred to on p. 7 l. 6 ? - Define term “self-efficacy”, might be considered differently in different disciplines Aim: The aim is a bit vague for me and might also not fully align to what you are planning:  - You state that your overall aim is to gain knowledge on women’s decision making for sustained participation in physical activity programs (p.6 l. 32/33, p. 8, l. 5 f.) – you are however then looking at women’s preferences. Considering the intention behavior gap it might be a bit ambitious to draw conclusions on the adherence to programs by asking for preferences. I recommend to be a bit more precise in the overall aim or discuss the possible discrepancy between preference and program adherence in your limitations. - Also, I do not fully understand p. 6 l. 39 ff. and alignment with your approach. You say that your study will evaluate the participatory approach and social components of the program but what you do is evaluating preferences for structural aspects of the program. Needs clarification. - Consider emphasizing that you will focus on structural components only p. 6 l.50 ff. - In general it was a bit difficult for me to find the section were you state the goal. I found it in several parts (p. 6 l. 39ff, p. 8 , p. 7l. 59, p. 8 l. 2) of the manuscript but one clearly identifiable section would be helpful, including clearly defined research questions /hypotheses - Consider carefully revising that you do not mix up the goals of (1) how you developed the choice task and (2) the goal for your experiment (I would only expect a goal for the latter in a study protocol, the development of the task was preparatory work for it) This also leads me to pointing out that I had the feelings that there are two large parts of your manuscript (which makes it a bit difficult to review the appropriateness of your study plan) The first one is the “Development of the choice task”. This work was already conducted and presents the methods and results for the development of the attributes and level included in the DCE, which I would consider preparatory work for your experiment (it proposes the tool you will use) However, this part is taking great part in your protocol and I am a bit confused how to evaluate this: 1) To stand alone as a qualitative results presentation it is not specific enough there are information and quotes missing (see my comments below). 2) To
--	--

	serve as an information only how the DCE was developed it is taking too much space in your manuscript (and great part of the abstract).  • The second part is the study protocol itself that describes how the DCE will be carried out and how it will be analysed  □ You keep jumping a bit back and forth between preparatory work and actual experiment, e.g. on page 12 l. 41ff. you present results of the preparatory work again “price was perceived and irrelevant attribute” even though you are already in the procedures of the experiment you plan to conduct. I would clearly distinguish between what was conceptual prework to develop your tool and the experiment itself. I think I would rather be more specific in the preparatory part and have two different parts of the manuscript or fully treat it as “preparatory work” in one (1) section of your manuscript (preferably in your methods section where you describe other measures e.g. health status etc. as well.) The following comment will now distinguish between these two bigger content areas. Development of DCE tool  - Please define conjoint analysis (p. 9 l.36) - The final list that you selected as researcher after “compilation of evidence” and presented to the BIG experts would be helpful in Annex, I was a bit surprised how the large list of attributes you present in Table A2 resulted in a quite decreased amount of attributes in Table 1. - Also for a qualitative approach it is necessary to know a bit more on the background of the “BIG experts”, are they professionals? Are they hired from the program? Or are they volunteers from the “communities” with the same socioeconomic characteristics as the participants? - How were the pretest participants selected? You state that these were volunteers, does this mean those who were most motivated? Please discuss possible limitations resulting from not selecting participant at random - What was the rationale for including women from two sides only? Please discuss possible limitations possibly resulting from not asking women from all sites and also from - Also you present results and new themes that resulted from the discussions with BIG experts and the think aloud study with participants (p. 10 l. 46ff., p. 11 l.3 ff., p. 12 l.41ff – I think if you want to mention the themes in such specific detail it is necessary that you integrate quotes for them, since quotes serve as evidence in qualitative results. Alternatively (and I guess I would prefer this) you may consider to be more general in the development of the tool or refer to another publication / OSF doc where you describe qualitative data in more detail. The way the qualitative results are presented here is somewhere in between too specific to be seen as preparatory work only and has too little detail for what one would expect when reading qualitative results. Study planned (Evaluation of preferences with DCE tool):  • Time line for the planned experiment is not clear, has the data been collected? On p. 8 l. 38 you mention time line May to December 2022 which is almost finished. • p. 13 l. 47 f.: please include reference that excluding opt-out options keep cognitive effort to minimum • Point 2.3.4 – the measures need to be described more specifically in order to evaluate its appropriateness, please consider including all questionnaires (sociodemographic aspects, health behavior, self-efficacy etc) in your Appendix
--	--

	 • On p. 15 l. 2 ff. you mention that you expect that 50% of women already left the program and you will only plan to include women remaining in the programs only (who, if I am understanding this correctly, seem to have a good adherence to the program because you mention that the programs started before COVID already). Wouldn't you (in particular) need the preferences from the women who dropped out the programs as well? Consider discussing why you decided not to or discuss as limitations if not possible (e.g. no contact info). • It would be helpful if you mentioned your research questions/ hypotheses (or expected results) for the experiment clearly • Data analyses: How will you handle missing data? How will you handle outliers? Which analyses program will you use? How will you analyze different classes of women? How will you measure education, health status, migration status? How will you treat all your variables (scale of measurement)? • Would you exclude any women from your analyses? E.g. women that do not represent your target population ("women in difficult life situation"), how do you define them? Other comments: - Something is wrong about the sentence p. 8 l. 50f. Please note that I was unable to attach files to this review (I would have uploaded my comments with better formatting).
--	---

REVIEWER	Stevens, Courtney Dartmouth-Hitchcock Medical Center, Psychiatry
REVIEW RETURNED	15-Dec-2022

GENERAL COMMENTS	In the strengths and limitations section on page 4, it's not clear how the first 3 points listed are either strengths or limitations of the study. These points read as statements about the study procedures without clear framing as strengths or limitations. Additionally, there appear to be grammatical errors in the second and third points listed, so it is unclear exactly what these two points are attempting to convey. Overall, more attention is needed to clearly describe study limitations. The manuscript would benefit from a careful proof read to reduce the number of minor grammatical errors (e.g.g., word choice errors, tense and lack of parallel structure, run-on sentences). Ultimately, these do not obstruct comprehension, but they are noticeable. There is lack of consistency in formatting the reference list; a number of citations are missing pertinent details about the publication source. Clinical trial registration information is not provided. Additionally, no SPIRIT checklist is provided. If these aspects of study reporting are not needed, please provide justification.
--

VERSION 1 – AUTHOR RESPONSE

Reviewer: 1
Dr. Vica Tomberge,
University of Bern

Comments to the Author:

Dear authors and editors,

I was delighted to be given the opportunity to review the study protocol of your highly relevant study which can hopefully contribute to the insight on the barriers and preferences of women from socially disadvantaged populations to engage in physical activity and other health promotion programs.

Please note that the line numbering was not consecutive. All comments refer to the pages assigned by

BMJ (not the page numbers on the authors' document, these differ).

First of all, I am not familiar with the DCE, thus I will limit myself to:

- giving feedback on the appropriateness of the conceptual development of the options included (based on contextual development of measures in general)
- but I can not give any feedback on the appropriateness of the choice set design (2.3.3. I. p. 13 22 ff.) or the analyses of it p. 14, I. 14 ff.

Abstract

1. Abstract contains shortforms which are not explained (BIG) and methodological approaches which are not self-explanatory and need a short definition (discrete choice experiment)

Please try to write the abstract sufficiently explicit to understand it on its own.

Thank you for pointing this out. We included an explanation of BIG and clearly explain that we will conduct a DCE "to investigate participant's preferences for a physical activity program".

2. Also your abstract contains large parts of describing preparatory work (development of tool) which is a bit confusing since this work was already conducted and is rather a tool for the experiment your plan, consider focusing on the experiment itself (see more specific comments on this below)

In preparing our protocol for a DCE to be published in BMJ Open, we referred to protocols that were already published in this journal and applied a similar structure to our paper. See for example:

<https://bmjopen.bmj.com/content/bmjopen/10/11/e043477.full.pdf>

<https://bmjopen.bmj.com/content/bmjopen/4/10/e006661.full.pdf>

<https://bmjopen.bmj.com/content/11/1/e042399.abstract>

In all these examples, it is common practice to present in the first part of the protocol a full account of the preparatory work on identification and selection of attributes, on the construction of experimental design and the choice sets.

Based on the content of previous protocols, we feel that it is an integral part of a DCE protocol to explain also preparatory work, and not just focusing on the experiment itself. Following the reviewer's comment, we clearly explain in the abstract that in this protocol we describe both the development of a DCE and how it will be conducted and analyzed. Furthermore, we included an explanatory section at

the beginning of section 2 (Methods) explaining the structure of the paper. Additionally, we clearly differentiated the preparatory stages described (as paragraph 2.3) from data analysis (paragraph 2.4).

We hope that these changes will help the reviewer and other readers to understand the structure of the paper.

Introduction /Theoretical background (p.5-6)

3. Your literature discussed socioeconomic barriers to health and health behavior exclusively, given that you are placing a women focused health program I would expect to read about the particular relevance of women as a target group and the intersection of gender and socioeconomic status related to health, health behavior and barriers.

Thank you for pointing out this very important point. We included a more precise explanation of the fact that women should be considered a vulnerable group for sedentary behavior in the introduction: "Furthermore, several reviews from different contexts have shown a gender gap in physical activity, demonstrating that women are more at risk of sedentary behavior than men because of several barriers, such as family responsibilities, safety concerns or lack of culturally appropriate sport opportunities. Therefore, the intersection of socioeconomic factors and sex specific factors makes women from disadvantaged socioeconomic backgrounds a vulnerable group at serious risk of

sedentary behavior.”

4. In line 45 p.5 you mention that the program is based on community-based participatory research, can you define/ elaborate this? In line 52 you then mention a “cooperative planning approach” which you define sufficiently, do you mean the same by this?

Thank you for pointing this out. We added a definition of community-based participatory research in the introduction: “The program is based on a community-based participatory research approach, i.e. a research paradigm that aims primarily at a closer communication and involvement of communities object of the investigation, to shape, manage and evaluate the intervention or the context studied.” More on CBPR can be read here: <https://academic.oup.com/ije/article/33/3/499/716600?login=false> As such, CBPR and “cooperative planning approach” are two related but slightly different concepts.

5. I am not convinced by the literature you cite (one position statement) for supporting that structural aspects, instead of individual-level determinants, might prove more effective to enhance physical activity. Since this is a big claim, convince me by adding experimental evidence or qualitative studies at least. P. 5 l. 30 ff. . Maybe it is not necessary to state that they are “more” effective, structural aspects are maybe simply one of many components of health behavior (besides individual cognitions etc).

Thank you for pointing this out. The reviewer is right in saying that the cited resource (a position paper from the European Association for the Study of Obesity) does not provide comparative evidence of downstream vs upstream interventions. We follow the suggestion of the reviewer and change the sentence accordingly: “While the majority of programs involve individual cognitive-behavioral approaches to enhance physical activity [12], it has been argued that a broader approach targeting also structural aspects might be necessary to effectively improve lifestyle [13].”

6. I need clarification on who you mean when you talk about communities, p.5 l. 50, is this a fixed group of people or a specific social group? Or is it the same as “sites” referred to on p. 7 l. 6 ?

Thank you for this comment. As it is customary in public health research, we use the term “community”

to refer to an undefined group of individuals who share the same environment and interact within that environment on the basis of shared interests and values. The term is used rather broadly in the literature on participatory research, although the community is identified as important level of health determinants and policy action. For this reason, some attempts were undertaken to define the concept

of community in a more precise way (see for example:

<https://www.ncbi.nlm.nih.gov/pmc/articles/PMC1446907/>). The term community does not coincide 1:1 with the term site, since the latter rather refers to a geographical location rather than the people living and interacting in it.

7. Define term “self-efficacy”, might be considered differently in different disciplines

For the measurement of self-efficacy, we used the ASKU scale (“Allgemeine Selbstwirksamkeit Kurzsкала”, i.e. Self-efficacy scale – short form), which is a standard instrument consisting of 3 questions to be used in socioeconomic investigations to measure subjective competence expectations.

You can find more here:

<https://www.testarchiv.eu/de/test/9006490#:~:text=In%20schwierigen%20Situationen%20kann%20ich,in%20der%20Regel%20gut%20l%C3%B6sen.>

<https://www.psycharchives.org/en/item/68dad76c-a94c-4f45-9158-2706a7447440>

The concept was cited in the introduction to explain the rationale and aims behind the BIG courses.

We are not currently planning to take this factor into account into our analysis, so we would prefer not to include a definition of this in the introduction, which would probably distract the reader and impair the flow.

Aim: The aim is a bit vague for me and might also not fully align to what you are planning:

8. You state that your overall aim is to gain knowledge on women’s decision making for sustained participation in physical activity programs (p.6 l. 32/33, p. 8, l. 5 f.) – you are

however then looking at women's preferences. Considering the intention behavior gap it might be a bit ambitious to draw conclusions on the adherence to programs by asking for preferences. I recommend to be a bit more precise in the overall aim or discuss the possible discrepancy between preference and program adherence in your limitations.

Thank you for pointing this out. Indeed, we are investigating women's preferences, which are only one

of many components of the decision-making process. We revised the paper accordingly, avoiding any reference to the decision-making process itself and rather stating that our aim is to understand preferences and reasons for participation.

9. Also, I do not fully understand p. 6 l. 39 ff. and alignment with your approach. You say that your study will evaluate the participatory approach and social components of the program but what you do is evaluating preferences for structural aspects of the program. Needs clarification.

Thank you for pointing this out. We reformulated the sentence to make the point clearer and more aligned with our aims:

"This DCE is the first one to evaluate the cooperative planning aspect and social activities component, as structural characteristics of a community-based participatory research program in the physical activity promotion field."

10. Consider emphasizing that you will focus on structural components only p. 6 l.50 ff.

Thank you for pointing this out. We included a reference to structural components in the abstract.

11. In general it was a bit difficult for me to find the section where you state the goal. I found it in several parts (p. 6 l. 39ff, p. 8 , p. 7l. 59, p. 8 l. 2) of the manuscript but one clearly identifiable section would be helpful, including clearly defined research questions

/hypotheses

Thank you for pointing this out. Indeed, we describe multiple aims in the protocol:

- the aims of the described DCE (last paragraph in the introduction)
- the aims of the BIG program (paragraph 2.1 "The BIG project")
- the aims of the NU-BIG evaluation project (paragraph 2.2 "The NU-BIG evaluation and the target group")

The last two items were explained as part of relevant context information. The aims of the DCE and of the protocol are explained in the last paragraph of the introduction. In order to make this clearer to the reader, we included a title for this paragraph ("Aims of this protocol and of the DCE"). We did not formulate any hypothesis since we used a rather exploratory approach.

12. Consider carefully revising that you do not mix up the goals of (1) how you developed the choice task and (2) the goal for your experiment (I would only expect a goal for the latter in a study protocol, the development of the task was preparatory work for it)

Please refer to our response for comment 2.

13. This also leads me to pointing out that I had the feelings that there are two large parts of your manuscript (which makes it a bit difficult to review the appropriateness of your study plan). The first one is the "Development of the choice task". This work was already conducted and presents the methods and results for the development of the attributes and level included in the DCE, which I would consider preparatory work for your experiment (it proposes the tool you will use).

However, this part is taking great part in your protocol and I am a bit confused how to evaluate this:

1) To stand alone as a qualitative results presentation it is not specific enough there are information and quotes missing (see my comments below).

2) To serve as an information only how the DCE was developed it is taking too much space in your manuscript (and great part of the abstract).

The second part is the study protocol itself that describes how the DCE will be carried out and how it will be analysed.

Please refer to our response for comment 2. We followed the structure and reported content of

previously published protocols for DCE studies. As the reviewer rightly points out, both the preparatory work for the DCE and the analysis plan are usually stated in a protocol. To make this structure clearer, we included a description of the structure of the paper at the beginning of section 2.

14. You keep jumping a bit back and forth between preparatory work and actual experiment, e.g. on page 12 l. 41ff. you present results of the preparatory work again “price was perceived and irrelevant attribute” even though you are already in the procedures of the experiment you plan to conduct. I would clearly distinguish between what was conceptual prework to develop your tool and the experiment itself. I think I would rather be more specific in the preparatory part and have two different parts of the manuscript or fully treat it as “preparatory work” in one (1) section of your manuscript (preferably in your methods section where you describe other measures e.g. health status etc. as well.)

The sentence that the reviewer is referring to belongs to the paragraph “pre-test with participants”. As explained in the protocol and as is good practice while planning a DCE experiment, the experiment (including the relevance of attributes and respective levels) should undergo a series of practical tests before roll-out of the study, including expert scrutiny and pre-tests with participants from the target group. While describing the preparatory steps for the DCE, we necessarily go “back and forth” since good practice foresees several rounds of checks and adjustments.

The sentence cited by the reviewer was extracted from a paragraph that explains the procedures and the results obtained in the pre-test with participants. Based on the feedback that we received from the participants included in this pre-test, we then optimized the DCE accordingly, again, as is good practice to do based on previous studies and protocols. Within the results of this part, we transparently

presented the result that the price range included in the test version was perceived as irrelevant from the test participants. Consequently, we increased the price range, as explained.

For the structure of the manuscript please refer to comment nr. 2.

The following comment will now distinguish between these two bigger content areas.

Development of DCE tool

15. Please define conjoint analysis (p. 9 l.36)

We included a definition of conjoint analysis in paragraph 2.3 (“The DCE is a type of conjoint analysis, i.e. a stated-preference method that involves an indirect comparison of choices, for example via ranking, rating or choice designs, to evaluate and quantify preferences for several attributes of an intervention.”)

16. The final list that you selected as researcher after “compilation of evidence” and presented to the BIG experts would be helpful in Annex, I was a bit surprised how the large list of attributes you present in Table A2 resulted in a quite decreased amount of attributes in Table 1.

Thank you for mentioning this. The process is indeed quite complex, these are the steps we followed and the attributes considered in each step:

- Compilation of evidence (literature search and program specific) led to the identification of several potential attributes (Appendix 2)
- Following the procedure described under the paragraph “Compilation of evidence (program specific) we narrowed down the long list to 10 potential attributes (most were excluded because they were present in the literature but were not applicable to the BIG context), reported in Appendix 3
- This list of 10 attributes was discussed with the experts in the next step: one additional attribute was identified and two were reframed (see Appendix 3)
- Further internal discussions described in Appendix 3, based on the insights from the expert meeting, led us to narrow down the attributes to 5 (reported in Table 1).

17. Also for a qualitative approach it is necessary to know a bit more on the background of the “BIG experts”, are they professionals? Are they hired from the program? Or are they

volunteers from the “communities” with the same socioeconomic characteristics as the participants?

Thank you for this question. As explained in the paragraph “consultations with experts”, they are “BIG course instructors, organizers or long-term participants, who actively took part in shaping BIG courses at their site and who are routinely interviewed and involved in the planning phases of the NU-BIG evaluation”.

We also added this clarification:

“The “expert group” of women was created at the beginning of the NU-BIG evaluation to implement the grounding principles of community-based participatory research and includes highly motivated women that are active in their sites in different roles. They receive a small compensation for their participation and are involved in multiple projects within the NU-BIG evaluation.”

18. How were the pretest participants selected? You state that these were volunteers, does this mean those who were most motivated? Please discuss possible limitations resulting from not selecting participant at random.

What was the rationale for including women from two sides only? Please discuss possible limitations possibly resulting from not asking women from all sites and also from [...?].

Thank you for bringing this up. We included the following sentence now in the paper: “We are aware that both samples selected for the expert interviews and the pre-test might not be representative of the targeted population of participating women. However, it was impossible for the study team to directly access participants from the study population (the study was planned before late spring 2022, when only selected courses were taking place due to Covid-19 restrictions). The potential bias induced

by this issue will be considered in the final evaluation.”

Some more info: For the pre-test, we asked the course coordinators and the expert group to recruit motivated participants that would be able to spare some time to fill in the questionnaire during a guided interview with one of the researchers (SP), against a small compensation. For this, we were able to recruit women from only two sites. We are aware of the potential bias that this selection might have had on the following steps of the DCE construction, but no other options were available to us in the limited time frame and during the pandemic situation.

19. Also you present results and new themes that resulted from the discussions with BIG experts and the think aloud study with participants (p. 10 l. 46ff., p. 11 l.3 ff., p. 12 l.41ff – I think if you want to mention the themes in such specific detail it is necessary that you integrate quotes for them, since quotes serve as evidence in qualitative results. Alternatively (and I guess I would prefer this) you may consider to be more general in the development of the tool or refer to another publication / OSF doc where you describe qualitative data in more detail. The way the qualitative results are presented here is somewhere in between too specific to be seen as preparatory work only and has too little detail for what one would expect when reading qualitative results.

We agree with the reviewer that this is an issue and thank her for bringing this up. To the best of our knowledge, there is no standard way of presenting qualitative analyses in the preparatory phase of a DCE. We feel that a detailed description including quotes from participants would exceed the scope of the paper, since this is not usual in other DCE protocols. Therefore, we opted for a more general, but anyway detailed description of the procedure, taking inspiration from other DCE protocols on the degree and type of details to be disclosed (see examples cited above).

Study planned (Evaluation of preferences with DCE tool):

20. Time line for the planned experiment is not clear, has the data been collected? On p. 8 l. 38 you mention time line May to December 2022 which is almost finished.

Thank you for pointing this out. Yes, data collection is almost finished. To my knowledge, the data is still being collected and will be collected throughout the first months of the year 2023. Therefore, we changed the timeline accordingly in the manuscript.

According to BMJ Open guidelines, the manuscript should be submitted before data collection is completed. Furthermore, “Reviewers will be instructed to review for clarity and sufficient detail. The

intention of peer review is not to alter the study design. Reviewers will be instructed to check that the study is scientifically credible and ethically sound in its scope and methods, and that there is sufficient detail to instil confidence that the study will be conducted and analysed properly” (from: <https://bmjopen.bmj.com/pages/authors>). So in our opinion, the nearing end of data collection should not be a problem. In any case, we will draw this point to the attention of the editor upon resubmission. 21. p. 13 l. 47 f.: please include reference that excluding opt-out options keep cognitive effort to minimum.

Thank you for pointing this out. The sentence points out a very simple statement, i.e. that adding choice options (in this case an opt-out option) will necessarily increase the cognitive burden for respondents (for example: answering question X requests less cognitive capacities than answering question X and Y). We do not intend for the sentence to be a general statement about opt-out options. Therefore, we reformulated the sentence as follows:

“To avoid increasing the cognitive effort for participants, we did not include any opt-out option.”

22. Point 2.3.4 – the measures need to be described more specifically in order to evaluate its appropriateness, please consider including all questionnaires (sociodemographic aspects, health behavior, self-efficacy etc) in your Appendix.

Unfortunately, we are not able to provide the full questionnaire to be attached in the appendix because some of the modules included are copyrighted. We agree with the reviewer that a full description and detailed account of the data collection instruments and methods for other variables is necessary, however we think that such full account will be more appropriate within the final publication.

23. On p. 15 l. 2 ff. you mention that you expect that 50% of women already left the program and you will only plan to include women remaining in the programs only (who, if I am understanding this correctly, seem to have a good adherence to the program because you mention that the programs started before COVID already). Wouldn't you (in particular) need the preferences from the women who dropped out the programs as well? Consider discussing why you decided not to or discuss as limitations if not possible (e.g. no contact info).

Thank you for pointing this out. The cited passage (“Even though assuming a 50% reduction in the sample, or in the response rate, the sample size is still higher than the minimum suggestion of 250.”) is intended as a conservative estimate and is to be interpreted as: even if we assume that half of the women that were participating before the Covid-19 pandemic dropped out, we can still very much reach the required sample size.

Collecting the preferences of women who dropped out of the program is indeed an extremely interesting question. However, no contact info of the women who dropped out was collected. The NUBIG questionnaire, as explained in paragraph 2.2, will be distributed to women who are participating

during data collection in the BIG courses. A limitation section is usually not included in a DCE protocol (see references under comment 2), but we agree with the reviewer that this aspect should be mentioned in the final manuscript.

24. It would be helpful if you mentioned your research questions/ hypotheses (or expected results) for the experiment clearly

The research question is mentioned in the last paragraph of the introduction. In order to clarify that, at this point, we cannot yet make any hypotheses on the results we included this sentence: “Besides a

clear preference for lower costs per course unit, a priori we cannot hypothesize which levels will be preferred by women.”

25. Data analyses: How will you handle missing data? How will you handle outliers? Which analyses program will you use? How will you analyze different classes of women? How will you measure education, health status, migration status? How will you treat all your variables (scale of measurement)? Would you exclude any women from your analyses? E.g. women that do not represent your target population (“women in difficult life situation”), how do

you define them?

Thank you for these questions. We provide answers for the following requests, with corresponding explanations in the paper:

- Analysis program used: we will use R-studio, we added a reference to this in the text.
- Analysis of different classes of women: as stated in the methods section, we will use latent class analysis to identify and characterize (using sociodemographic characteristics) clusters of women with heterogeneous preferences (see paragraph 2.4).
- Measure/scale of other items: we included a new table (Table 2) in the paper explaining details of used instruments for the sociodemographic information collected and used in the DCE analysis.
- Exclusion criteria, missing data and outliers: We added two sentences to the data analysis section that described which women will be excluded based on the number of completed choices-sets. Then, the analysis is carried out according to available cases. Imputation of missing information is uncommon for DCEs. The basic assumption of DCEs is to elicit preferences which are representative for the sample under consideration. The preferences will be estimated by logistic regression models and thus, outliers will be captured by the regression model resulting in estimates of available data.
- Definition of target group: the target group of the BIG program are women in difficult life situations, a definition of which is given in paragraph 2.1. However, the BIG physical activity courses are open to every woman who would like to participate. Since the target group of the questionnaires are women who participate in BIG courses (paragraph 2.2), the actual population targeted by the DCE will include all women participating (without a distinction whether they belong to the target group of BIG or not). The extent to which the surveyed sample actually belongs to the intended target group of BIG will be investigated in the descriptives of the paper. We included a reference to this in paragraph 2.3.5

Other comments:

26. Something is wrong about the sentence p. 8 l. 50f.

Thank you for pointing this out, we proof-read the whole manuscript as suggested by reviewer 2.

Reviewer: 2

Dr. Courtney Stevens,
Dartmouth-Hitchcock Medical Center

Comments to the Author:

27. In the strengths and limitations section on page 4, it's not clear how the first 3 points listed are either strengths or limitations of the study. These points read as statements about the study procedures without clear framing as strengths or limitations. Additionally, there appear to be grammatical errors in the second and third points listed, so it is unclear exactly what these two points are attempting to convey. Overall, more attention is needed to clearly describe study limitations.

Thank you for pointing this out. We reformulated the strengths and limitations following the reviewer's comment.

28. The manuscript would benefit from a careful proof read to reduce the number of minor grammatical errors (e.g.g., word choice errors, tense and lack of parallel structure, run-on sentences). Ultimately, these do not obstruct comprehension, but they are noticeable.

There is lack of consistency in formatting the reference list; a number of citations are missing pertinent details about the publication source.

Thank you for this comment. We revised and proofread the whole manuscript.

29. Clinical trial registration information is not provided. Additionally, no SPIRIT checklist is provided. If these aspects of study reporting are not needed, please provide justification.

Thank you for pointing this out. The described DCE is an economic experiment, for which no register (similar to clinical trial registers) exists. A registration in a clinical trial register would also not be appropriate from our perspective because of the nature of the questions included (not targeting

primarily health and behavior but individual preferences). The SPIRIT checklist is a checklist for clinical

trials protocols and as such, is not appropriate for a DCE.

The only checklist that we know that fits our purpose is the Bridges et al. (2011) checklist for conjoint analysis applications in healthcare:

[https://www.valueinhealthjournal.com/article/S1098-3015\(10\)00083-](https://www.valueinhealthjournal.com/article/S1098-3015(10)00083-5/fulltext?_returnURL=https%3A%2F%2Flinkinghub.elsevier.com%2Fretrieve%2Fpii%2FS109830151000835%3Fshowall%3Dtrue)

[5/fulltext?_returnURL=https%3A%2F%2Flinkinghub.elsevier.com%2Fretrieve%2Fpii%2FS1098301510](https://www.valueinhealthjournal.com/article/S1098-3015(10)00083-5/fulltext?_returnURL=https%3A%2F%2Flinkinghub.elsevier.com%2Fretrieve%2Fpii%2FS109830151000835%3Fshowall%3Dtrue)

[000835%3Fshowall%3Dtrue](https://www.valueinhealthjournal.com/article/S1098-3015(10)00083-5/fulltext?_returnURL=https%3A%2F%2Flinkinghub.elsevier.com%2Fretrieve%2Fpii%2FS109830151000835%3Fshowall%3Dtrue)

The checklist is applicable to studies reporting not only the DCE methodology but also the results so that in our case only a small part of the items could be responded at this time (protocol). Therefore, we included in the paper a reference to the checklist, with a commitment to make use of the checklist once the final paper (with results) will be published.

VERSION 2 – REVIEW

REVIEWER	Stevens, Courtney Dartmouth-Hitchcock Medical Center, Psychiatry
REVIEW RETURNED	06-Apr-2023
GENERAL COMMENTS	The authors took care to address feedback from the first review and this version is much improved.